# Trypanosoma cruzi PARP is enriched in the nucleolus and is present in a thread connecting nuclei during mitosis

**María Laura Kevorkian**[1], **Salomé C. Vilchez Larrea**[1,2], **Silvia H. Fernández Villamil**[1,3]*

**1** Instituto de Investigaciones en Ingeniería Genética y Biología Molecular "Dr. Héctor N. Torres", Consejo Nacional de Investigaciones Científicas y Técnicas, Ciudad Autónoma de Buenos Aires, Argentina, **2** Departamento de Fisiología, Biología Molecular y Celular, Facultad de Ciencias Exactas y Naturales, Universidad de Buenos Aires, Ciudad Autónoma de Buenos Aires, Argentina, **3** Departamento de Química Biológica, Facultad de Farmacia y Bioquímica, Universidad de Buenos Aires, Ciudad Autónoma de Buenos Aires, Argentina

☯ These authors contributed equally to this work.

* silvia.villamil@gmail.com, s.villamil@ingebi.conicet.gov.ar

## Abstract

Poly (ADP-ribose) polymerase (PARP) is responsible for the synthesis of ADP-ribose polymers, which are involved in a wide range of cellular processes such as preservation of genome integrity, DNA damage signaling and repair, molecular switches between distinct cell death pathways, and cell cycle progression. Previously, we demonstrated that the only PARP present in *T. cruzi* migrates to the nucleus upon genotoxic stimulus. In this work, we identify the N-terminal domain as being sufficient for TcPARP nuclear localization and describe for the first time that TcPARP is enriched in the parasite's nucleolus. We also describe that TcPARP is present in a thread-like structure that connects two dividing nuclei and co-localizes with nucleolar material and microtubules. Furthermore, ADP-ribose polymers could also be detected in this thread during mitosis. These findings represent a first approach to new potential TcPARP functions inside the nucleus and will help understand its role well beyond the largely described DNA damage response protein in trypanosomatids.

## Introduction

Poly (ADP-ribose) polymerases (PARPs) catalyze the transfer of ADP-ribose units from NAD$^+$ to a target protein or an elongating polymer chain, a post-translational modification known as PARylation [1]. Recent studies have shown that nucleic acids can also serve as substrates of reversible ADP-ribosylation [2]. Enzymes able to carry out PARylation have been described in all types of eukaryotic organisms, from mammals and plants to lower eukaryotes, except for yeasts. Orthologs have also been found in bacteria [3]. In humans, the PARP family consists of 18 members grouped into subfamilies. One of these subfamilies, the "DNA-dependent" PARPs (PARP-1, PARP-2 and PARP-3), contains domains capable of recognizing DNA strand breaks, thus enabling the recruitment of proteins involved in DNA repairing [4–6]. The most extensively studied function of human PARP-1 (hPARP-1) is its role in DNA damage signaling and repair. Nevertheless, over the last years, the functions of PARPs have

**Funding:** Agencia Nacional de Promoción Científica y Tecnológica (ANPCyT), Argentina, PICT 2020-00476.

**Competing interests:** The authors have declared that no competing interests exist.

transcended from being solely associated with genomic damage to being involved in diverse processes such as the regulation of transcription factors, protein and lipid metabolism, immune response, chromatin remodeling, and cell death. Free poly(ADP ribose) (PAR) also mediates pathophysiological processes such as stroke, neuroinflammation and neurodegeneration [5, 7–10].

Our research team has pioneered the study of PAR metabolism in trypanosomatids, with particular focus on *Trypanosoma cruzi*, the agent responsible for Chagas´ disease, an anthropozoonosis from the American continent that has recently spread to other countries [11–17]. Understanding the biology of this parasite is needed to improve epidemiological control as well as therapeutic methods for the treatment of this neglected disease. *T. cruzi* and *Trypanosoma brucei*, the causal agent of Sleeping sickness in humans and Nagana in cattle, possess only one PARP (TcPARP and TbPARP, respectively) and PARG, in contrast to the large family described in higher eukaryotes [17]. Immunohistochemical studies revealed the extranuclear distribution of trypanosomatid PARP in basal conditions and its ability to shuttle from this space into the nucleus after a genotoxic stimulus [14, 16]. This mechanism permits that the only PARP present in these organisms can be strategically delivered to the kinetoplast as well as to the nucleus, to potentially regulate both nuclear and mitochondrial DNA (kDNA) metabolism. Preservation of the integrity of kDNA is crucial in *T. cruzi* since it represents between 20 to 25% of the total DNA and encodes important proteins [17]. Akin PARP from other species, DNA breaks induce TcPARP activation, leading to nuclear PAR production [14].

We found that the single 65 kDa PARP in *T. cruzi*, named TcPARP, is expressed in the three developmental stages of the parasite: epimastigotes present in the insect vector, and trypomastigotes and amastigotes that reside in the mammalian host [12]. TcPARP shares some structural features with DNA-dependent PARPs, as three of their typical domains are present in the parasite's enzyme. TcPARP contains a WGR domain, named after the conserved tryptophan, glycine and arginine residues, which has a nucleotide-binding function. A central regulatory domain and catalytic domain are also present, but TcPARP lacks the typical zinc-finger N-terminal domain, involved in the DNA regulation of human PARP-1 activity [17]. Nevertheless, we described an N-terminal region rich in basic amino acids, also present in *T. brucei* PARP, which is a DNA binding site required for DNA-dependent enzymatic activation [12, 16]. In hPARP-1 and hPARP-2, a similar basic amino acid domain has been demonstrated to bind to DNA and cooperate with their corresponding catalytic domain [4].

In the Trypanosomatidae family, an early divergent eukaryote group, closed mitosis occurs and the nucleolus presents some important differences compared to human. Nucleologenesis is described as a continuous process without the concentration of nucleolar material within intermediate nuclear bodies. Therefore the direct passage of nucleolar components at the end of mitosis has been suggested [18, 19]. Interestingly, in *T. cruzi* the presence of a thread connecting the two dividing nuclei has been reported [18].

In this work, we identify the N-terminal domain as being sufficient for TcPARP nuclear localization and describe for the first time that TcPARP is enriched in the parasite´s nucleolus. We also report that both TcPARP and PAR are present in a thread that connects dividing nuclei and co-localizes with a nucleolar marker. These results could extend PAR roles well beyond the largely described DNA damage response in trypanosomatids.

## Materials and methods

### Parasite cultures

*Trypanosoma cruzi* epimastigotes (CL Brener strain) were cultured at 28˚C in liver infusion tryptose (LIT) medium (5 g.l$^{-1}$ liver infusion, 5 g.l$^{-1}$ bacto-tryptose, 68 mM NaCl, 5.3 mM KCl,

22 mM $Na_2HPO_4$, 0.2% (w/v) glucose and 0.002% (w/v) hemin) supplemented with 10% (v/v) fetal bovine serum, 100 $U.ml^{-1}$ penicillin and 100 $mg.l^{-1}$ streptomycin. For transgenic and wild type parasites growth curves, epimastigotes in the exponential growth phase were diluted to $6.10^6$ parasites $ml^{-1}$ placed in 96-well sterile plates in 100 μl aliquots. Optical density ($OD_{600nm}$) of the cultures was determined each day for five days to follow cell viability. All conditions were tested in triplicates.

## Plasmid constructions

Several sets of primers were used for the amplification of different TcPARP protein domain combinations by PCR using a plasmid bearing a copy of *T. cruzi* PARP gene (GenBank: DQ061295) as a template [12]. The primers used in the amplification reaction of the different constructs were: FL-Fw 5'-CCA TGG CAC CAA AGA AGT TAT CAG GAG C-3' and FL-Rv 5'-GTC GAC TGA TAT TTA AAC CCC ACA TGG ACC A-3' for TcPARP-FL; ΔN-Fw 5'-GAA TCC ATG GTG TAC GAA AAA GGC-3' and ΔN-Rv 5'-CTC GAG TCA AGC GTA ATC TGG AAC ATC GTA TGG GTA ATG ATA TTT AAA CCC-3' for TcPARPΔN; ΔNW-Fw 5'-GAA TCC ATG GGA GCG GCG GAG GAC GAG-3' and ΔNW-Rv 5' CTC GAG TCA AGC GTA ATC TGG AAC ATC GTA TGG GTA ATG ATA TTT AAA CCC-3' for TcPARPΔNW; ΔRC-Fw 5'-GAA TCC ATG TCA CCA AAG AAG TTA TCA GGA-3' and ΔRC-Rv 5'- CTC GAG TCA AGC GTA ATC TGG AAC ATC GTA TGG GTA ATA ATC AAC GTA CAT CAA-3' for TcPARPΔRC; ΔWRC-Fw 5'-GAA TTC ATG GTG TAC GAA AAA GGC-3' and ΔWRC-Rv 5'-CTC GAG TCA AGC GTA ATC TGG AAC ATC GTA TGG GTA ATA ATC AAT GTC CAT CAA-3' for TcPARPΔWRC; ΔNRC-Fw 5'-GAA TTC ATG GTG TAC GAA AAA GGC-3' and ΔNRC-Rv 5'-CTC GAG TCA AGC GTA ATC TGG AAC ATC GTA TGG GTA ATA ATC AAC GTA CAT CAA-3' for TcPARPΔNRC. PCR products were cloned into a pGEM-T Easy vector and subcloned into a pRIBOTEX vector to overexpress the different TcPARP constructs tagged with HA. TcPARP full-length was cloned into pTEX-GFP vector. *T. cruzi* epimastigotes of the CL Brener strain were transfected with the above-mentioned construct as previously described [14].

Stable cell lines were achieved after 60 days of treatment with 500 μg $ml^{-1}$ of G418. Full-length TcPARP encompasses from amino acids 1 to 592; ΔN from 127 to 592; ΔNW from 212 to 592; ΔRC from 1 to 211; ΔWRC from 1 to 126 and ΔNRC from 127 to 211. The recombinant proteins were overexpressed in *T. cruzi* epimastigotes and the presence of chimeric proteins was confirmed by Western Blot (S1A Fig). In some cases other bands different than the expected ones were observed. PARP undergoes several post-translational modifications, of which the most important one is self-PARilation. We thought this is likely the case of TcPARP-FL, which contains all its domains. In that case, the upper band, which appears regularly in these Western Blots, would be attributed to modified TcPARP-FL. We believe that, in the case of ΔNW and ΔWRC, partial degradation or incomplete transcription could have generated bands of slightly smaller sizes.

Growth curves of transgenic parasites showed no significant differences compared to wild type epimastigotes, both in basal condition or after 200 μM $H_2O_2$ treatment for 10 minutes, (S1B and S1C Fig). These results showed that TcPARP constructs did not alter normal epimastigote growth neither under basal nor genotoxic stress conditions and, thus, had no deleterious effect over TcPARP activity under DNA damage. However, since the expression of the constructs cannot be controlled, we cannot rule out that the lack of phenotype could be due to different expression levels of the chimeras. Growth curves of the different strain of epimastigotes are shown as relative $OD_{600nm}$ at each day ($OD\ t_x$) to the initial value ($OD\ t_0$) vs day of reading.

Error bars show the SEM of five independent experiments each one of them in technical triplicates. A one-way ANOVA revealed no significant differences between the different growth curves.

### SDS–PAGE and Western blot analysis

Whole cell lysates from transgenic epimastigotes were quantified using the Bradford method. Proteins (40 μg) were run on 10% SDS–PAGE gel and transferred to an Amersham Hybond-ECL nitrocellulose membrane (GE healthcare), according to the manufacturer's instructions. Prestained Protein Ladder–Blue Plus® IV (10–180 kDa) TransGen Biotech Co., LTD was used as protein marker. Immunodetection of TcPARP constructs was carried out using a 1:1000 dilution of rat polyclonal antibody directed against HA, influenza virus hemagglutinin (Roche), followed by 1:4000 anti-rat horseradish peroxidase (HRP) conjugated antibody. For full-length TcPARP-GFP, anti-GFP (B-2): sc-9996 (Santa Cruz Biotechnology) antibody was used, followed by 1:4000 anti-mouse horseradish peroxidase (HRP) conjugated antibody The signal was detected with the Western Lightning Plus-ECL kit (PerkinElmer).

### Immunofluorescence

For immunolocalization of TcPARP, transgenic *T. cruzi* epimastigotes ($10^7$ parasites ml$^{-1}$) were treated with 200 μM $H_2O_2$ for 10 min, fixed for 20 min with 4% (w/v) paraformaldehyde in PBS, permeabilized with fresh PBS– 0.1% Triton X-100 and blocked for 1 h at room temperature with 3% (w/v) BSA in PBS. For full-length TcPARP, an anti-GFP antibody 1:500 was used and TcPARP constructs were detected with 1:500 rat polyclonal antibody directed against HA, followed by 1:500 Alexa Fluor 488 goat anti-rat IgG (Sigma-Aldrich) conjugated antibody. The nucleolus was identified using 1:100 L1C6 monoclonal antibody and mouse monoclonal anti α-Tubulin (Sigma-Aldrich) for α-Tubulin. Alexa Fluor 594 goat anti-mouse IgG (Sigma-Aldrich) conjugated antibody was used as a secondary antibody. The PAR binding reagent used was rabbit polyclonal anti PAR (BD Pharmingen™). Alexa Fluor 594 goat anti-rabbit IgG (Sigma-Aldrich) conjugated antibody was used as a secondary antibody. Nuclei were stained with 2 μg ml$^{-1}$ DAPI (Sigma) in PBS. Coverslips were mounted with VectaShield® and then visualized using an Olympus BX41/FV300 (S2–S5 Figs) or Leica SPE confocal microscope (Figs 1–4). Samples incubated in the absence of the primary antibody showed no unspecific binding of secondary antibodies (S2 Fig). All images in each series of at least two independent experiments were taken with the same setting at the same microscopy session. If modified, all were subjected to the same degree of brightness/contrast adjustment and Gaussian blur filtering, including the control without a primary antibody. The ImageJ free software was used for confocal image processing and JACoP Fiji's plugin was used for colocalization analysis.

## Results

### TcPARP is enriched in the nucleolus of epimastigotes

As already known from previous research, subcellular localization of TcPARP is altered in response to genotoxic stress, which stimulates TcPARP to re-localize from the cytoplasm and accumulate in the nucleus in the epimastigote form of the parasite [14]. Constructs with different arrangements of TcPARP domains were designed and the resulting recombinant proteins were overexpressed in *T. cruzi* epimastigotes (S1 Fig). Actively growing cells were examined for the distribution of the recombinant proteins after treatment with 200 μM $H_2O_2$ for 10 minutes (S3 Fig), which has been shown to induce nuclear translocation of endogenous TcPARP

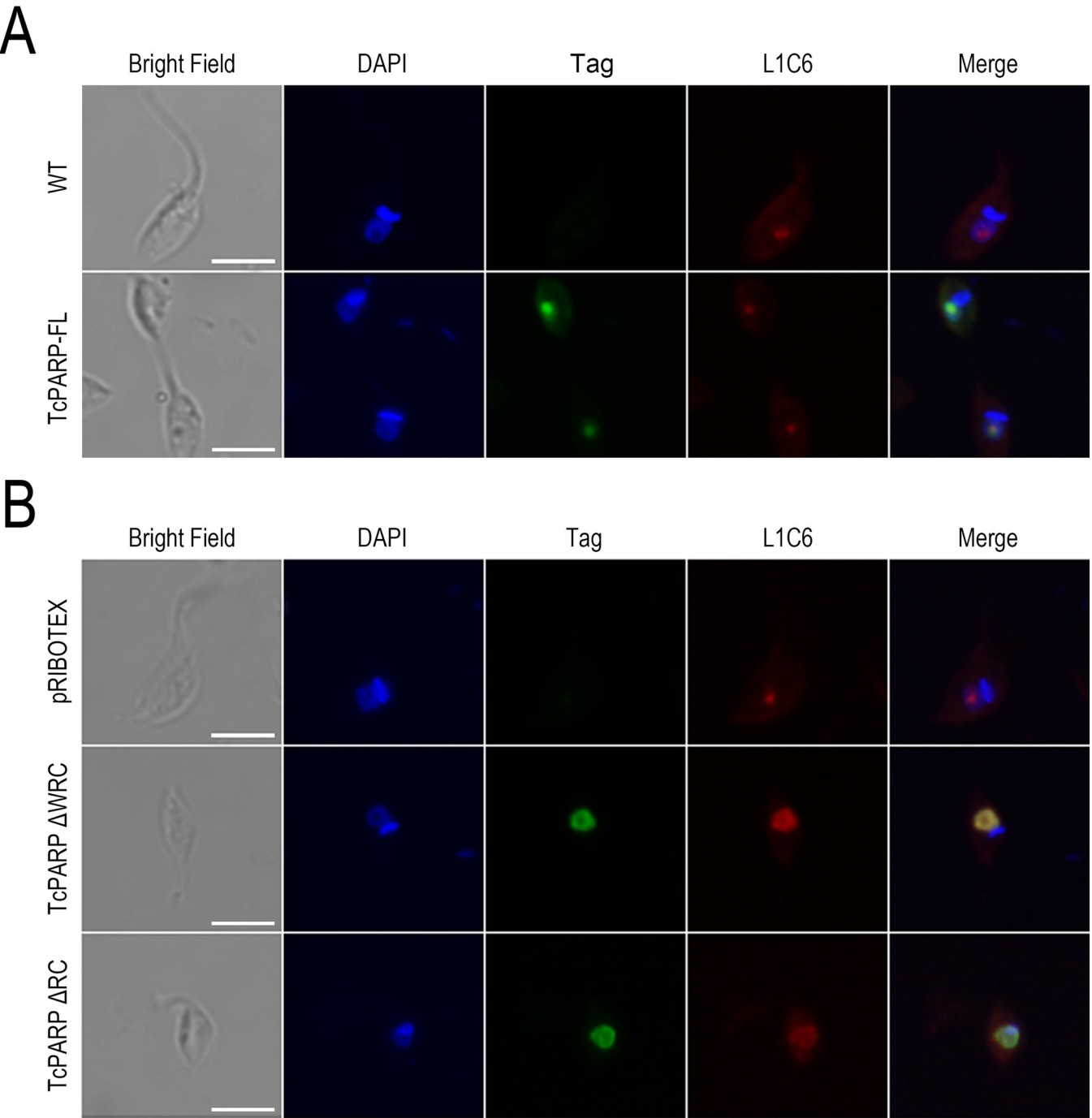

**Fig 1. Colocalization of TcPARP constructs with nucleolus marker L1C6.** (A) Indirect immunofluorescence (IF) of wild type or TcPARP-FL overexpressing epimastigotes. (B) IF of epimastigotes overexpressing the empty vector pRIBOTEX, TcPARPΔWRC or TcPARPΔRC. In both cases, detection of the nucleolar region was performed using an antibody directed against L1C6 and the TcPARP construct with the corresponding anti-tag antibody. Scale bar: 5 μm. Pearson Correlation Coefficients (PCC), which ranges from −1 to +1 (from anti-colocalization to complete colocalization), were determined: 0.85; 0.93 and 0.61 for TcPARP-FL, TcPARPΔWRC and TcPARPΔRC respectively.

[12]. Nuclear localization was observed for full length TcPARP and the constructs bearing the N-terminal domain, under control conditions or after $H_2O_2$ treatment. However, TcPARP constructs which lacked the N-terminal domain did not localize to this organelle even after

genotoxic stress, showing a scattered pattern throughout the cytoplasm (S3 Fig). These results indicate that the N-terminal domain is necessary for the nuclear localization of the protein.

Particularly, nucleolar enrichment of the recombinant proteins that bear the N-terminal domain (TcPARPΔWRC, TcPARPΔRC and TcPARP-FL) in overexpressing epimastigotes could be observed. A higher intranuclear fluorescence density of the anti-tag signal was detected within the area with reduced DAPI staining (S4 Fig). This region corresponds to the nucleolus, which has low DNA content [20]. In order to confirm this localization in transgenic epimastigotes, a colocalization assay was carried out using the nucleolus marker L1C6, which recognizes a nucleolar antigen both in *T. brucei* and *T. cruzi* [21]. Fig 1A shows the location of L1C6 in the nucleolar structure of CL Brener wild type epimastigotes. The signal corresponding to TcPARP-FL overlaps with the labeling of L1C6 inside the nucleus of the parasites, which asserts the subnuclear localization of this protein. The same phenomenon could be observed for truncated TcPARP constructs, including the N-terminal domain, where the fluorescent signal colocalizes with nucleolus marker L1C6. The intensity correlation analysis confirmed the nucleolar localization of TcPARP proteins (Fig 1). We noticed that in some parasites, both TcPARP (full length or ΔWRC truncated version) and L1C6 nucleolar marker were also found in the nucleoplasm (27% ΔWRC-overexpressing parasites showed nucleolar localization vs nucleoplasm localization (n = 125), whereas 51% TcPARP-FL-expressing parasites showed nucleolar localization (n = 159) vs nucleoplasm localization). In *Drosophila* it has been reported that PARP-1 depletion resulted in the mislocalization of nucleolar specific proteins and loss of nucleolar structure [22]. To determine whether TcPARP enzymatic activity and the presence of PAR are essential for maintaining nucleolar integrity in *T. cruzi*, we inhibited TcPARP activity by adding the $NAD^+$ analogue, 3 aminobenzamide (3AB) or the more selective TcPARP inhibitor, Olaparib [23]. We did not detect de-localization of the nucleolar marker in the presence PARP inhibitors, ruling out this hypothesis (S5 Fig). Another possibility would be that the high transcriptional activity generates the de-localization of nucleolar marker. To test this hypothesis, we overexpressed a non-nuclear protein such as PI3K TcVps34 in epimastigotes, both in absence or in the presence of cycloheximide. We did not observe significant differences in any of the parasites overexpressing recombinant proteins in comparison to the control (S5 Fig). Dispersion of the nucleolar antigen L1C6 has also been described by Gluenz et al. and attributed to epimastigotes in stationary phase [20], but since we worked with cultures during exponential growth, the reason for the nucleolar marker de-localization in our models remains unknown.

## TcPARP is present in a connecting thread between nuclei of dividing epimastigotes

*T. cruzi* epimastigotes undergoing the cellular division process show the presence of a thread connecting the two dividing nuclei, which has been previously associated to nucleolar proteins [18]. We confirmed colocalization of TcPARP with the nucleolar specific marker L1C6 in this narrow link between both nuclei by indirect immunofluorescence. The presence of L1C6 could be identified in the entire structure of the thread between nuclei, colocalizing with the truncated TcPARPΔWRC version of TcPARP or the complete protein (Fig 2). Colocalization of TcPARP with tubulin in the connecting thread is shown in Fig 3, which implies the presence of TcPARP in the mitotic spindle during nuclear segregation and brings forward possible new functions for TcPARP that need further study.

## PARylated proteins are present in the connecting thread

We further studied whether TcPARP was catalytically active in the thread. To do this, we evaluated the presence of PAR, product of PARP activity, associated with the thread. The presence

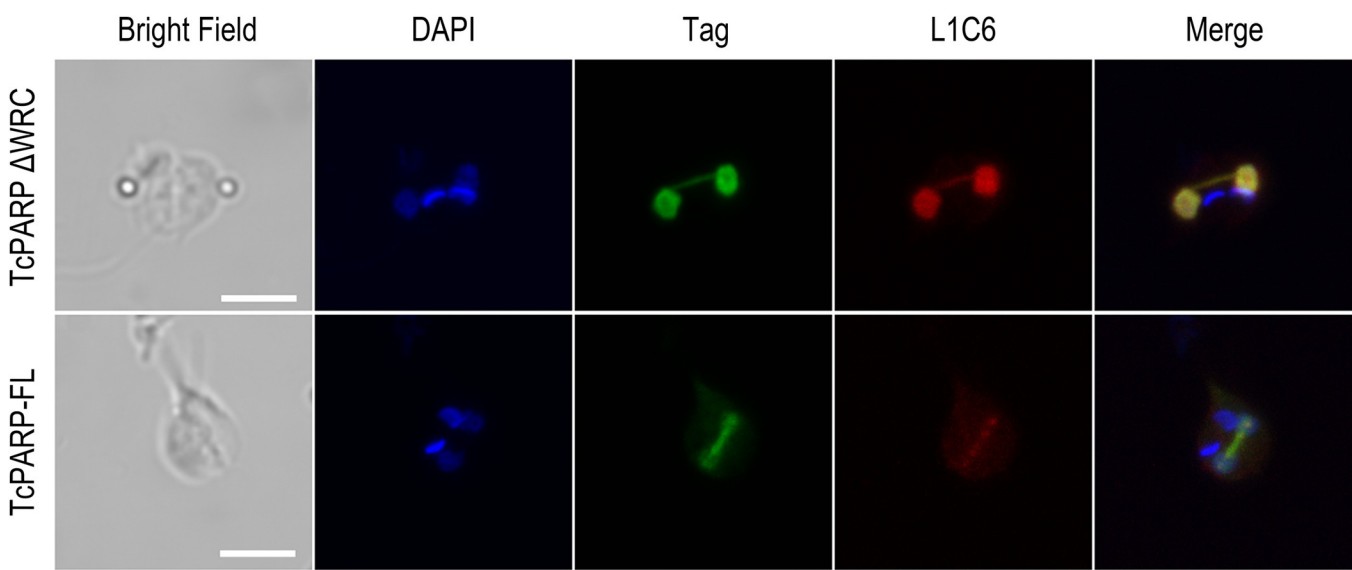

**Fig 2. Colocalization of TcPARP with nucleolar material in the connecting thread.** Indirect immunofluorescence of epimastigotes overexpressing TcPARP-FL or the N-terminal domain of TcPARP (TcPARPΔWRC). Nucleolar proteins were evidenced with antibody directed against L1C6. Recombinant TcPARP-FL and chimeric peptide were detected with the corresponding anti-tag antibody. Scale bar: 5 μm. Pearson Correlation Coefficients were: 0.84 and 0.91 for TcPARP-FL and TcPARPΔWRC respectively.

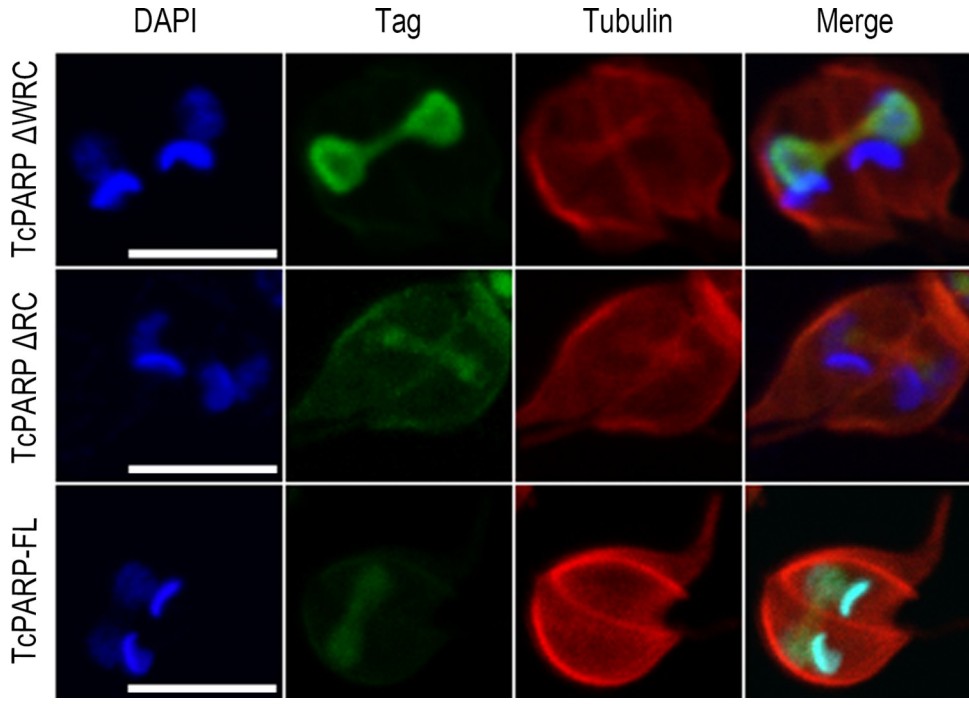

**Fig 3. Colocalization of TcPARP with tubulin in the connecting thread.** Indirect immunofluorescence of epimastigotes overexpressing TcPARP-FL or the N-terminal domain of TcPARP (TcPARPΔWRC and TcPARPΔRC). Tubulin was detected by an antibody directed against α-Tubulin and recombinant variants of TcPARP with the corresponding anti-tag antibody. Scale bar: 5 μm.

of PAR was assessed in epimastigotes that overexpressed the complete TcPARP or the domains responsible for its nuclear localization. Basal levels of PAR are low and therefore hardly detectable by the methods commonly employed [12]. Therefore, we subjected the parasites to a genotoxic stimulus in order to enhance PAR production. Fig 4 shows the presence of PAR in the

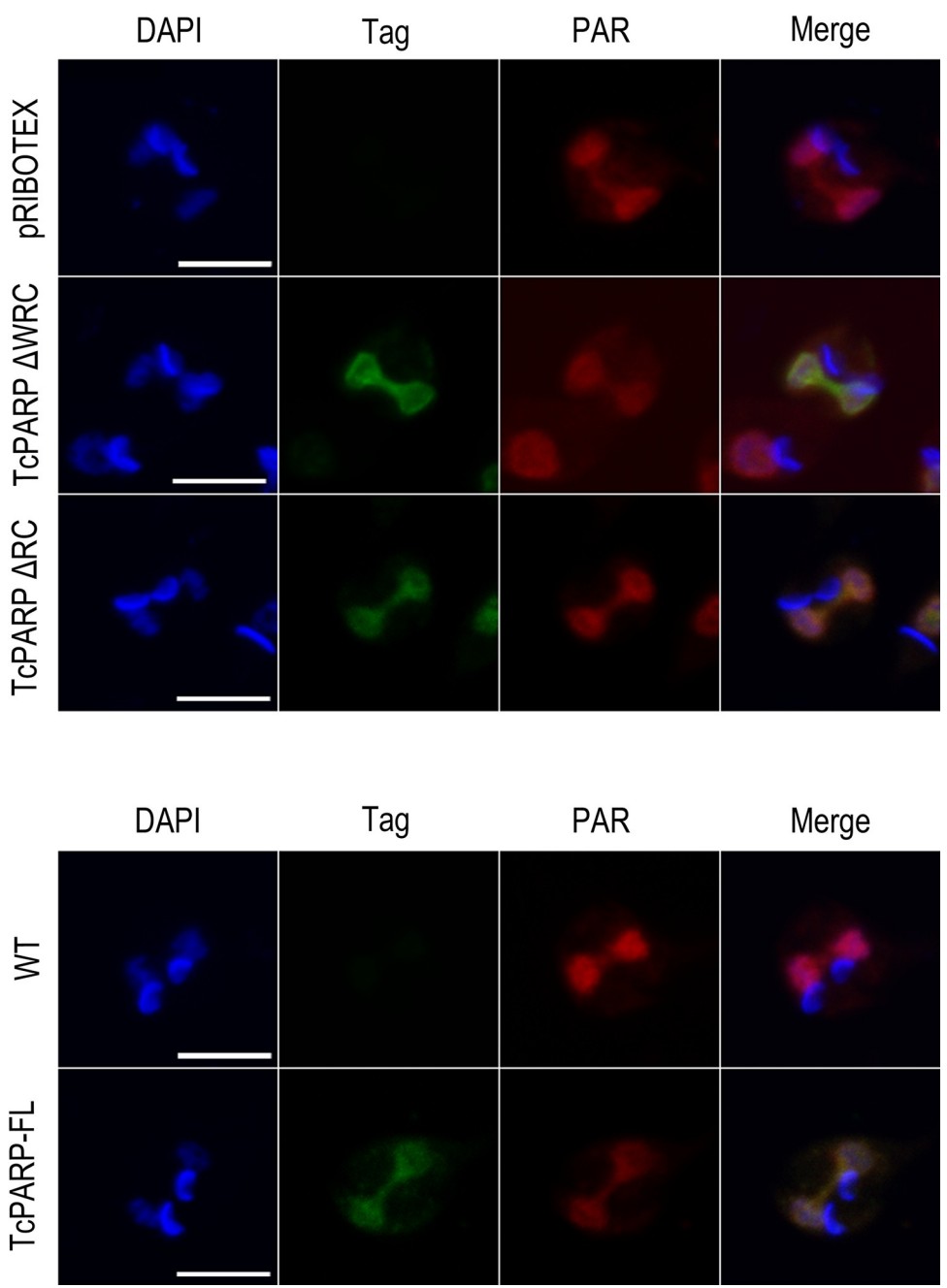

**Fig 4. Presence of PAR in the connecting thread between nuclei of dividing epimastigotes.** Upper panel: Indirect immunofluorescence (IF) of epimastigotes bearing the empty vector pRIBOTEX or overexpressing the TcPARPΔWRC or TcPARPΔRC constructs, after 200 μM $H_2O_2$ treatment for 10 minutes. Lower panel: IF of wild type epimastigotes or TcPARP-FL overexpressing parasites after 200 μM $H_2O_2$ treatment for 10 minutes. In both cases, PAR was detected using a polyclonal antibody directed against PAR chains and recombinant variants of TcPARP with the corresponding anti-tag antibody. Scale bar: 5 μm. Pearson Correlation Coefficients were: 0.98, 0.72 and 0.97 for TcPARP-FL, TcPARPΔWRC and TcPARPΔRC respectively.

nucleus of $H_2O_2$-treated parasites; PAR colocalizing with the TcPARP constructs is highlighted in the thread that connects the dividing nuclei. PAR localization showed the same pattern observed in epimastigotes expressing the empty vector pRIBOTEX, or in wild type parasites (Fig 4). This demonstrates that PARylated proteins are present in this structure as a consequence of endogenous TcPARP activity.

## Discussion

In *Trypanosoma cruzi*, TcPARP translocates to the nucleus under genotoxic stress and in this work we have demonstrated that the N-terminal domain is sufficient for this translocation. Despite no canonical NLS has been found, roughly 30% of N-terminal amino acids are basic [14], which indicates the possible presence of a nuclear localization signal within this region. Interestingly, the TcPARP N-terminal contains six repeats of the amino acid sequence AAKKA, which is also present in *T. cruzi* histone H2A and in all H1, and which could mediate the DNA binding necessary for TcPARP functions. A *T. cruzi* importin α with the classical structure of ARM repeats was recently reported [24]. This protein was functional as a nuclear transport factor in these parasites, supporting the idea that this import machinery was an early acquisition of the eukaryotic cell. Numerous authors have described the subnuclear localization of PARP-1 in *Drosophila* and in mammalian cells [22, 25–28], where up to 40% of PARP-1 is enriched in the nucleolus. Our finding that TcPARP localizes in the nucleolus of *T. cruzi* epimastigotes is consistent with that data. Altogether, our results demonstrate that the N-terminal domain of TcPARP is involved in nuclear and nucleolar targeting.

In *Trypanosoma cruzi* closed mitosis occurs, in which the chromatin does not condense and the nucleolus is not clearly identified. The nucleolar structure persists throughout closed mitosis, where the nucleolus is elongated and, finally, divided into two structures [29]. Observation of trypanosomes expressing various fusion proteins allowed visualization of mitotic cells connected by a thin thread [30]. In dividing cells, we found that TcPARP was located in the connecting thread that separates the daughter nuclei, colocalizing with tubulin and the nucleolar marker. As determined by DAPI staining, nuclear DNA seems to be excluded from the thread indicating a late stage in mitosis. This observation implies that TcPARP, like other nucleolar components, forms part of the connecting thread after chromosomal segregation. Among other PARP functions in mammalian nucleolus, Raemaekers (2003) identified the modification of NuSAP1 (nucleolar protein associated with spindle 1), a protein involved in the mitotic spindle formation, which concentrates in the nucleolus. In metaphase and early anaphase, NuSAP is redistributed, colocalizing with and stabilizing the spindle microtubules [31]. Both NuSAP and tubulin are targets of PARylation [32]. Similarly, NUMA1 (nuclear mitotic apparatus protein 1), which participates in the clustering of microtubules into poles as a prerequisite for bipolar spindle organization, is associated with PAR. Its PARylation and PAR-binding property may promote correct assembly of bipolar spindles by crosslinking NUMA1 molecules between spindle poles [33]. Thus, the PARylation of spindle proteins associated with microtubules plays an important role in the assembly and function of the mitotic spindle, where PARP could work as a Microtubules Associated Protein. Spindle associated proteins (SAP) NuSAP1-3 were described in *T. brucei* and depletion of these proteins resulted in mitotic defects [34]. NuSAP 1 and 2 are also annotated as putative proteins for *T. cruzi* (TcCLB.508821.20 and TcCLB.509353.50). In addition to this function, PAR has been postulated as a structural component of the spindle that contributes to creating the force exerted by microtubules to elongate the spindle and separate chromosomes [35]. Furthermore, ECT2 (epithelial cell trans-forming sequence 2 oncogene), necessary for the control of cytokinesis, is recruited by PARylated α-tubulin to the spindle during metaphase as a prerequisite for

functional cytokinesis and completion of mitosis [36]. Then, the nucleolar location of TcPARP and associated PARylated proteins during mitosis could be involved in spindle formation, not only through the PARylation of nucleolar proteins but also as a result of their structural functions. The Aurora B Kinase forms the chromosomal passenger complex playing essential roles in mitosis and becomes highly PARylated in response to DNA damage [37]. A functional analogue of Aurora B Kinase, named Tc AUK1 was recently identified in *T. cruzi* with a possible role in the initiation of kinetoplast duplication [38].

The localization of TcPARP in the nucleolus and mitotic structures indicates that this protein could be performing a wide array of new roles, leaving aside the notion that this enzyme is functional only in the nucleus in response to genotoxic damage. The possible participation of TcPARP in the nucleolus as an enzyme related to the formation of the mitotic spindle requires deeper investigation.

## Supporting information

**S1 Fig. (A)** Constructs bearing different combinations of TcPARP domains expressed in transgenic epimastigotes. Left Panel: Schematic Diagram of truncated TcPARP recombinant proteins made through the combination of different domains and fused to HA tag. TcPARP full length was tagged with GFP. Numbers under each diagram indicate amino acid positions. Theoretical molecular weight was calculated considering the molecular weight of HA (1 kDa) or GFP (27 kDa). Right panel: Western Blot on total extract of *T. cruzi* epimastigotes transfected with plasmids bearing the indicated constructs, using the appropriate antibody against the tag. The arrows indicate the expected molecular weight corresponding to the expression of the different constructs. LC: Ponceau-stained membrane as loading control. Parasite growth curves under basal or oxidative stress condition. Epimastigotes of transgenic and wild type (WT) lines in exponential growth phase were incubated in basal condition **(B)** or in the presence of 200 μM $H_2O_2$ for 10 minutes **(C)**. After treatments cells were collected by centrifugation at 3000g/5min at room temperature, washed with PBS, suspended in LIT ($6.10^6$ parasites $ml^{-1}$) and placed in 96-well plates in 100 μl aliquots. The $H_2O_2$ concentration used was sublethal and permitted parasite growth. All the growth curve tests were carried out in three or more independent experiments, each one of them in technical triplicates. Results are expressed as relative growth, OD $_{600nm}$ at each day was normalized to the initial value (OD $t_x$/OD $t_0$). One-way ANOVA, show no significant differences compared to the wild type. (TIF)

**S2 Fig. Immunofluorescence images control.** Epimastigotes were incubated without primary antibodies. Alexa Fluor 488 goat anti-rat IgG conjugated antibody 1:500 (upper panel) and Alexa Fluor 594 goat anti-mouse IgG conjugated antibody 1:500 (lower panel) were used as a secondary antibodies. Blue represents DAPI staining of kinetoplast and nuclear DNA. Samples incubation in the absence of the primary antibody showed no unspecific binding of secondary antibodies. Scale bar: 5 μm. (TIF)

**S3 Fig. Subcellular localization of TcPARP constructs.** Indirect immunofluorescence of *T. cruzi* epimastigotes (CL Brener strain) that overexpress TcPARP-FL or different combinations of protein domains, under basal conditions (Control) or treated with 200 μM hydrogen peroxide ($H_2O_2$) for 10 minutes. (A) TcPARP-FL. (B) TcPARPΔN. (C) TcPARPΔNW. (D) TcPARPΔRC. (E) TcPARPΔWRC. (F) TcPARPΔNRC. Scale bar: 10 μm. (TIF)

**S4 Fig. Subnuclear localization of TcPARP-FL, TcPARPΔRC and TcPARPΔWRC.** Indirect immunofluorescence of *T. cruzi* epimastigotes (CL Brener strain) that overexpress TcPARP-FL, TcPARPΔRC or TcPARPΔWRC. Fluorescence intensity profile for each channel (Blue or Green) was analyzed using ImageJ software on the line shown in the DAPI panel. (TIF)

**S5 Fig. Dispersion of the nucleolus.** Actively growing wild type epimastigotes preincubated for 1 h in the presence of the $NAD^+$ analogue, 3 aminobenzamide (3AB), or the TcPARP inhibitor, Olaparib; and PI3K TcVps34 overexpressing parasites (TcVps34 overexpression) in the presence or absence of 100 μg.ml$^{-1}$ cycloheximide (CHX), were fixed and labeled with L1C6 antibody. Scale bar: 5 μm. (TIF)

## Acknowledgments

The authors are grateful to Dr. Daniel Sanchez, Instituto de Investigaciones Biotecnológicas (IIB), Universidad Nacional de San Martín–CONICET, for providing L1C6 antibody. We are also grateful to Dr. Alejandra Schoijet, Instituto de Investigaciones en Ingeniería Genética y Biología Molecular "Dr. Héctor N. Torres", for the PI3K TcVps34 epimastigotes.

## Author Contributions

**Conceptualization:** María Laura Kevorkian, Salomé C. Vilchez Larrea, Silvia H. Fernández Villamil.

**Data curation:** María Laura Kevorkian, Salomé C. Vilchez Larrea, Silvia H. Fernández Villamil.

**Formal analysis:** María Laura Kevorkian, Salomé C. Vilchez Larrea, Silvia H. Fernández Villamil.

**Investigation:** María Laura Kevorkian, Salomé C. Vilchez Larrea, Silvia H. Fernández Villamil.

**Methodology:** Salomé C. Vilchez Larrea, Silvia H. Fernández Villamil.

**Project administration:** Silvia H. Fernández Villamil.

**Supervision:** Salomé C. Vilchez Larrea, Silvia H. Fernández Villamil.

**Writing – original draft:** Silvia H. Fernández Villamil.

**Writing – review & editing:** María Laura Kevorkian, Salomé C. Vilchez Larrea, Silvia H. Fernández Villamil.

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
