## [Decision Letter · Decision Letter 0]

10 May 2022

PONE-D-22-09375*Trypanosoma cruzi* PARP is enriched in the nucleolus and is present in a thread connecting nuclei during mitosis.PLOS ONE

Dear Dr. FERNANDEZ,

Thank you for submitting your manuscript to PLOS ONE. After careful review, we feel that it has merit but does not fully meet the criteria for publication in PLOS ONE in its current form. Therefore, we invite you to submit a revised version of the manuscript that addresses all the points raised by both reviewers..

Please perform quantifications of your images as suggested by reviewer #2, also indicate how many times you performed each experiment and carefully complete the method section, make sure all experiments/approaches are present and well described.

Please pay attention to the comments "Lack of reports regarding data/experiments" .  Reviewer #2 send you many useful comments that will improve your manuscript, please take the time to answer all of them.

We look forward to receiving your revised manuscript.

Kind regards,

Claude Prigent

Academic Editor

PLOS ONE

Journal Requirements:

- https://www.cambridge.org/core/journals/microscopy-and-microanalysis/article/abs/nucleologenesis-in-trypanosoma-cruzi/1D537BAAFF5DD76FC04E13C85D41377C

The text that needs to be addressed involves lines 68-73 in your Introduction.

In your revision ensure you cite all your sources (including your own works), and quote or rephrase any duplicated text outside the methods section. Further consideration is dependent on these concerns being addressed.

Reviewers' comments:

Reviewer's Responses to Questions

**Comments to the Author**

1. Is the manuscript technically sound, and do the data support the conclusions?

Reviewer #1: Yes

Reviewer #2: Partly

2. Has the statistical analysis been performed appropriately and rigorously? 

Reviewer #1: Yes

Reviewer #2: I Don't Know

3. Have the authors made all data underlying the findings in their manuscript fully available?

Reviewer #1: Yes

Reviewer #2: Yes

4. Is the manuscript presented in an intelligible fashion and written in standard English?

Reviewer #1: Yes

Reviewer #2: Yes

5. Review Comments to the Author

Reviewer #1: I really enjoyed reading the manuscript of Kevorkian et al on PARP in T. cruzi. In my opinion, the authors presented a comprehensive story definitely worth publishing. I have one suggestion and one cosmetic critique.

Suggestion: please extend info on PARP in other trypanosomatids

Cosmetic critique: please fix references for consistency and compliance with PLoS One style (word capitalization)

Reviewer #2: See uploaded document for additional comments.

I thank the authors for their work, which I enjoyed reading.

I think overall the study shows promise and will open up new interest into PARPs in NTDs, but I feel there is a lack of quantitative data to support conclusions as most of the work is observational at this point.

That said, many of the core issues could be addressed by re-analysing images and clarifying aspects of certain experiments which I feel would vastly, and quickly improve this dataset.

6. PLOS authors have the option to publish the peer review history of their article (what does this mean?). If published, this will include your full peer review and any attached files.

Reviewer #1: **Yes: **Vyacheslav Yurchenko

Reviewer #2: No

---

## [Author Response · Author response to Decision Letter 0]

16 Sep 2022

We are very grateful for the editor´s and reviewers' comments and believe they will greatly improve the manuscript. We are sending the manuscript including both the reviewers' and editor's recommendations. Quantifications of the images and number of experiments are now indicated. We have also carefully completed the introduction and methods section as suggested and responded point by point the reviewers´ comments in the rebuttal letter (see Rebuttal letter please)

Manuscript meets PLOS ONE's style requirements and overlapping text was rewritten and rephrased according to the editor suggestion.

We thank the possibility of improving our work and hope that the manuscript would now be suitable for publication in PLoS One.

---

## [Decision Letter · Decision Letter 1]

18 Oct 2022

PONE-D-22-09375R1*Trypanosoma cruzi* PARP is enriched in the nucleolus and is present in a thread connecting nuclei during mitosis.PLOS ONE

Dear Dr. FERNANDEZ,

Thank you for submitting your manuscript to PLOS ONE. After careful consideration, we feel that it has merit but does not fully meet PLOS ONE’s publication criteria as it currently stands. Therefore, we invite you to submit a revised version of the manuscript that addresses the points raised during the review process. Please check reviewer2 comments, which seem relatively minor to me.

We look forward to receiving your revised manuscript.

Kind regards,

Claude Prigent

Academic Editor

PLOS ONE

Journal Requirements:

Reviewers' comments:

Reviewer's Responses to Questions

**Comments to the Author**

1. If the authors have adequately addressed your comments raised in a previous round of review and you feel that this manuscript is now acceptable for publication, you may indicate that here to bypass the “Comments to the Author” section, enter your conflict of interest statement in the “Confidential to Editor” section, and submit your "Accept" recommendation.

Reviewer #1: All comments have been addressed

Reviewer #2: (No Response)

2. Is the manuscript technically sound, and do the data support the conclusions?

Reviewer #1: Yes

Reviewer #2: Yes

3. Has the statistical analysis been performed appropriately and rigorously? 

Reviewer #1: Yes

Reviewer #2: Yes

4. Have the authors made all data underlying the findings in their manuscript fully available?

Reviewer #1: Yes

Reviewer #2: Yes

5. Is the manuscript presented in an intelligible fashion and written in standard English?

Reviewer #1: Yes

Reviewer #2: Yes

6. Review Comments to the Author

Reviewer #1: All my concerns have been properly addressed. I think the revised version of the manuscript is an improvement over R0.

Reviewer #2: Reviewer 2 comments:

This reviewer wishes to thank the authors for their improvements to the manuscript -

the article has been significantly improved to accommodate suggestions by both reviewers.

1) Figure 1 in manuscript

I personally think it would be worth including the data under stress just so people are aware that there is no effect under these conditions – would be fine as a supplementary figure if preferred.

2) Also, why not use the inhibitor to look at spindle formation/cytokinetic defeats etc? Why not carefully profile the spindle progression across mitosis and the formation of the thread structure?

Reviewer response to author’s comment:

Suggestions that may help, though not necessary for this particular study. Perhaps the authors could consider employing expansion microscopy to help visualise the spindle thread better or even TEM could assist. Perhaps FISH of the telomeres to help define mitosis?

Whilst I do also completely agree with the authors it is challenging in parasites of this size, I also work exclusively on kinetoplastids and know it is very possible to examine the spindle and more miniature structures in this organism.

There are papers tracking spindle dynamics/cytokinesis etc in Leishmania and T. brucei. For example, Ambit et al. 2011 is a good example of this type of analysis (though in Leishmania) https://journals.asm.org/doi/10.1128/EC.05118-11

I also appreciate that defining the mitotic stages is not easy but perhaps looking through single cell transcriptomics/ cell cycle specific transcriptomics studies (re cell cycle) may help find potential markers of different cell cycle regulated proteins which could be tagged, for example. Though lacking in T. cruzi, profiles from related parasites would no doubt be useful in guiding some decisions.

Minor point 1:

Overall, regarding controls – it is fine to keep the main figure clear, but I would still put the corresponding controls in a supplementary figure to reassure readers that they have been done. Just because it doesn’t look pretty or fit neatly in a figure doesn’t mean they do not belong in a manuscript. Thus I would strong urge the authors to include all controls performed.

Even an image of a ponceau stained gel would be helpful. The comparison is not part of the questions, I understand, but if you already did the western blot then it would have been possible to do this without additional work. For additional bands on a western blot – perhaps indicating you are aware of them, and you are unsure of what they maybe would be helpful.

Reviewer comment:

What if you are not seeing phenotypes simply due to poor overexpression of one truncated piece Vs another – looks like from the WB that the expression of the recombinant version of TcPARP is different in each cell line.

Reviewer response to author’s comments:

Though the authors are saying that it is not the focus of the study,

you are comparing how cells have grown relative to WT without any indication of how well your experimental system is performing. As said by the authors – you cannot control the level of expression, so it is possible that comparing the different construct expressing strains to the WT is not showing a phenotype because that construct is not expressing the protein at a high enough level when compared to the WT level?

Perhaps worth considering this as a possibility irrespective of the focus of the paper for future works.

Minor points 2:

Line 62: have shown only one PARP and PARG ---- perhaps “posses (?) only one PARP and PARG”

Line 63 – reference for this statement

Lines 91-97 – please add reference (s) for these statements

Line 235: Did or did not (sorry in my version it reads “did observe significant…” but I presume you mean didn’t!

Line 241: I would suggest “but since we worked with cultures during exponential growth, the reason for the nucleolar marker de- localization in our model remains unknown. “

7. PLOS authors have the option to publish the peer review history of their article (what does this mean?). If published, this will include your full peer review and any attached files.

Reviewer #1: No

Reviewer #2: No

---

## [Author Response · Author response to Decision Letter 1]

29 Nov 2022

Dear Editor

We are very grateful for the reviewers’ comments and we are sending the manuscript attending both the reviewers´ and editor’s recommendations. 

Reviewer #1: All my concerns have been properly addressed. I think the revised version of the manuscript is an improvement over R0.

We would like to thank the referee for reviewing the manuscript and for the nice comments on our work. 

Reviewer #2: 

Reviewer 2 comments:

This reviewer wishes to thank the authors for their improvements to the manuscript -

the article has been significantly improved to accommodate suggestions by both reviewers.

We would like to thank reviewer 2 once more for the critical reading of our manuscript and for his/her help in improving the here revised version. 

1) Figure 1 in manuscript

I personally think it would be worth including the data under stress just so people are aware that there is no effect under these conditions – would be fine as a supplementary figure if preferred. 

In figures S1B and S1C we have included growth curves of all strains used in this work both in basal conditions and under stress, as was suggested. 

2) Also, why not use the inhibitor to look at spindle formation/cytokinetic defeats etc? Why not carefully profile the spindle progression across mitosis and the formation of the thread structure?

Reviewer response to author’s comment:

Suggestions that may help, though not necessary for this particular study. Perhaps the authors could consider employing expansion microscopy to help visualise the spindle thread better or even TEM could assist. Perhaps FISH of the telomeres to help define mitosis?

Whilst I do also completely agree with the authors it is challenging in parasites of this size, I also work exclusively on kinetoplastids and know it is very possible to examine the spindle and more miniature structures in this organism.

There are papers tracking spindle dynamics/cytokinesis etc in Leishmania and T. brucei. For example, Ambit et al. 2011 is a good example of this type of analysis (though in Leishmania) https://journals.asm.org/doi/10.1128/EC.05118-11

I also appreciate that defining the mitotic stages is not easy but perhaps looking through single cell transcriptomics/ cell cycle specific transcriptomics studies (re cell cycle) may help find potential markers of different cell cycle regulated proteins which could be tagged, for example. Though lacking in T. cruzi, profiles from related parasites would no doubt be useful in guiding some decisions.

We would really like to thank the reviewer for the suggestions and for sharing his/her experience on the subject. We have read the recommended paper and images are outstanding. We believe that all this information will be very important to continue with our work in order to make progress in this project. 

Minor point 1:

Overall, regarding controls – it is fine to keep the main figure clear, but I would still put the corresponding controls in a supplementary figure to reassure readers that they have been done. Just because it doesn’t look pretty or fit neatly in a figure doesn’t mean they do not belong in a manuscript. Thus I would strong urge the authors to include all controls performed. 

Even an image of a ponceau stained gel would be helpful. The comparison is not part of the questions, I understand, but if you already did the western blot then it would have been possible to do this without additional work. For additional bands on a western blot – perhaps indicating you are aware of them, and you are unsure of what they maybe would be helpful.

All controls are within the manuscript either in the figures or in the supplementary material as required. Controls for immunofluorescences were added as Supplementary figure 2. Images of Ponceau-stained membrane were added as loading control for Western blot in figure S1A, as was suggested.

As recommended, in the description of figure S1A we have inserted a paragraph where we refer to the possible reasons behind the additional bands for some constructs in the Western blot. Lines 136-142 in the new manuscript.

Reviewer comment:

What if you are not seeing phenotypes simply due to poor overexpression of one truncated piece Vs another – looks like from the WB that the expression of the recombinant version of TcPARP is different in each cell line.

Reviewer response to author’s comments:

Though the authors are saying that it is not the focus of the study,

you are comparing how cells have grown relative to WT without any indication of how well your experimental system is performing. As said by the authors – you cannot control the level of expression, so it is possible that comparing the different construct expressing strains to the WT is not showing a phenotype because that construct is not expressing the protein at a high enough level when compared to the WT level?

Perhaps worth considering this as a possibility irrespective of the focus of the paper for future works.

The reviewer's comment is very pertinent and we will take it into account for future works. We thought it was important to clarify this statement to the readers, so a small paragraph was inserted in the new version of the manuscript. Lines 143-153 in the new manuscript.

Minor points 2:

Line 62: have shown only one PARP and PARG ---- perhaps “posses (?) only one PARP and PARG” 

Was corrected.

Line 63 – reference for this statement 

Reference was added.

Lines 91-97 – please add reference (s) for these statements

References were added.

Line 235: Did or did not (sorry in my version it reads “did observe significant…” but I presume you mean didn’t! 

We apologize for the mistake, this was corrected.

Line 241: I would suggest “but since we worked with cultures during exponential growth, the reason for the nucleolar marker de- localization in our model remains unknown. “ 

Sentence was corrected as suggested.

---

## [Editor Report · Decision Letter 2]

13 Dec 2022

*Trypanosoma cruzi* PARP is enriched in the nucleolus and is present in a thread connecting nuclei during mitosis.

PONE-D-22-09375R2

Dear Dr. FERNANDEZ,

We’re pleased to inform you that your manuscript has been judged scientifically suitable for publication and will be formally accepted for publication once it meets all outstanding technical requirements.

Kind regards,

Claude Prigent

Academic Editor

PLOS ONE
---

## [Editor Report · Acceptance letter]

20 Dec 2022

PONE-D-22-09375R2 

*Trypanosoma cruzi* PARP is enriched in the nucleolus and is present in a thread connecting nuclei during mitosis. 

Dear Dr. FERNANDEZ:

I'm pleased to inform you that your manuscript has been deemed suitable for publication in PLOS ONE. Congratulations! Your manuscript is now with our production department. 

Kind regards, 

on behalf of

Dr. Claude Prigent 

Academic Editor

PLOS ONE